# Metagenomic Analysis of Plasma Microbial Extracellular Vesicles in Patients Receiving Mechanical Ventilation: A Pilot Study

**DOI:** 10.3390/jpm12040564

**Published:** 2022-04-02

**Authors:** Jinkyeong Park, Jae Jun Lee, Yoonki Hong, Hochan Seo, Tae-Seop Shin, Ji Young Hong

**Affiliations:** 1Department of Pulmonary and Critical Care Medicine, Kyung Hee University Hospital at Gangdong, Kyung Hee University School of Medicine, Seoul 05278, Korea; pjk3318@gmail.com; 2Institute of New Frontier Research Team, Hallym University College of Medicine, Chuncheon 24253, Korea; iloveu59@hallym.or.kr; 3Department of Internal Medicine, School of Medicine, Kangwon National University, Kangwon National University Hospital, Chuncheon 24289, Korea; h-doctor@hanmail.net; 4MD Healthcare Inc., Seoul 03293, Korea; hcseo@mdhc.kr (H.S.); tsshin@mdhc.kr (T.-S.S.); 5Division of Pulmonary and Critical Care Medicine, Department of Medicine, Chuncheon Sacred Heart Hospital, Hallym University Medical Center, Chuncheon 24253, Korea

**Keywords:** extracellular vesicle, microbiome, pneumonia, ICU, biomarkers

## Abstract

Background: Previous studies reported a significant association between pneumonia outcome and the respiratory microbiome. There is increasing interest in the roles of bacterial extracellular vesicles (EVs) in various diseases. We studied the composition and function of microbiota-derived EVs in the plasma of patients receiving mechanical ventilation to evaluate whether they can be used as a diagnostic marker and to predict clinical outcomes. Methods: Plasma samples (*n* = 111) from 59 mechanically ventilated patients (41 in the pneumonia group; 24 in the nursing home and hospital-associated infection [NHAI] group) were prospectively collected on days one and seven. After isolating the bacterial EVs from plasma samples, nucleic acid was extracted for 16S rRNA gene pyrosequencing. The samples were evaluated to determine the α and β diversity, bacterial composition, and predicted functions. Results: Principal coordinates analysis revealed significantly different clustering of microbial EVs between the pneumonia and non-pneumonia groups. The proportions of *Lactobacillus*, *Cutibacterium*, and *Sphingomonas* were significantly different between the pneumonia and non-pneumonia groups. In addition, the abundances of *Lactobacillus* and *Bifidobacterium* were significantly higher in the non-NHAI than the NHAI group. In the analysis of β diversity, the structure of microbial EVs differed significantly different between 28-day survivors and non-survivors (Bray-Curtis distance, *p* = 0.014). Functional profiling revealed significant differences between the pneumonia and non-pneumonia groups. The longitudinal change in predicted functions of microbial EV genes showed a significant difference between 28-day survivors and non-survivors. Conclusions: Bacterial microbiota–derived EVs in the plasma have potential as diagnostic and prognostic markers for patients receiving mechanical ventilation. Further large prospective studies are needed to determine the clinical utility of plasma microbiota-EVs in intubated patients.

## 1. Introduction

Bacterial microbiota-derived extracellular vesicles (EVs) play an important role in intracellular communication [1]. EVs contain DNA, proteins, mRNAs, and microRNAs that regulate inflammation and immune responses [2]. Pathogen-derived EVs are associated with the development of respiratory diseases such as chronic obstructive pulmonary disease, asthma, and neutrophilic pulmonary inflammation [3,4,5].

Shukla et al. reported that gut microbiota-derived EVs affect the lung immune response through EV component transfer in the gut-lung axis [6]. Gut microbiota-derived EVs can travel through the systemic circulation to reach many organs, such as the lungs, resulting in immune and metabolic responses.

EVs produced by commensal bacteria promote mucosal tolerance to protect the host, while EVs derived from dysbiosis of the microbiota contribute to the development of lung diseases [7]. For example, EVs produced by *Streptococcus pneumonia* perform immunomodulatory functions to evade the humoral host defense [8].

Pneumonia is a leading cause of hospitalization and death. Severe pneumonia may necessitate intensive care unit (ICU) admission and invasive mechanical ventilation [9,10]. The microbial etiology of pneumonia depends on the immune status of patients and the location of disease [11]. The new culture-independent next-generation sequencing methods have several advantages over conventional culture-based analysis, including higher sensitivity and a shorter turnaround time) [12]. Notably, microbial EVs (MEVs) in the blood have not been studied previously. In particular, evaluating the MEV metagenome may elucidate the systematic effects of bacterial infection and their potential as novel biomarkers of patient risk and clinical outcomes [13].

We hypothesized that the composition of bacterial EVs in plasma may reflect the presence of pneumonia, the risk of nursing home and hospital-associated infection (NHAI), and 28-day mortality. We analyzed the bacterial composition and predicted function of MEV genes in the plasma of mechanically ventilated patients with or without pneumonia.

## 2. Materials and Methods

### 2.1. Study Population and Plasma Sample Collection

This prospective study included 59 patients (41 with pneumonia and 18 without pneumonia) who received mechanical ventilation within 48 h of admission to the ICU of Chuncheon Sacred Heart Hospital, South Korea, between July 2017 and August 2018. We excluded patients who were younger than 18 years, received mechanical ventilation >48 h after ICU admission or for <7 days, or had a history of neuromuscular disease, such as amyotrophic lateral sclerosis.

Pneumonia was diagnosed on the basis of symptoms and treated in accordance with international guidelines [14,15]. The risk for NHAI was considered increased if the patient was a nursing home resident with poor functional status and/or had a recent hospitalization (within 90 days) or recent antibiotic use (within 180 days) [16].

Clinical data, including demographic characteristics, indications for intubation, the presence of acute respiratory distress syndrome (ARDS), severity of illness (Acute Physiology and Chronic Health Evaluation [APACHE] II and Sequential Organ Failure Assessment [SOFA] scores), and the Glasgow Coma Scale (GCS) were recorded. The 28-day all-cause mortality, in-hospital mortality, and duration of mechanical ventilation were the outcomes of interest. Study participants were divided into survivors and non-survivors based on the 28-day all-cause mortality

This study was approved by the Institutional Review Board of Chuncheon Sacred Heart hospital (approval no. 2017-47). Written informed consent was obtained from the participants and the study was conducted in accordance with approved guidelines.

### 2.2. EV Isolation and DNA Extraction from Human Plasma

Plasma samples were obtained one and seven days after initiation of mechanical ventilation. Plasma samples were collected into a BD Vacutainer (BD, Franklin Lakes, NJ, USA) and centrifuged at 2000× *g* for 15 min at 4 °C. The EVs were isolated from plasma samples by performing differential centrifugation using a microcentrifuge (Labogene 1730R; BMS, Seoul, Korea) at 10,000× *g* for 10 min at 4 °C. The supernatant was filtered through a 0.22-μm filter to eliminate bacteria and foreign particles. Next, the EVs were boiled for 40 min at 100 °C and centrifuged for 30 min at 13,000 rpm at 4 °C. EV DNA was extracted using a DNeasy PowerSoil Kit (Qiagen, Hilden, Germany) and quantified using the QIAxpert system (Qiagen). This was performed with the same protocol as previously described [17]. The EVs were characterized by transmission electron microscopy, nanoparticle tracking analysis and Dynamic Light Scattering.

### 2.3. Amplicon Sequencing Using EV DNA from Human Plasma Samples and Data Processing

The extracted DNA was amplified using primers targeting the 16S V3 and V4 hypervariable regions of the prokaryotic 16S rRNA gene. The primers used were 16S_V3_F (5′-TCGTCGGCAGCGTCAGATGTGTATAAGAGACAGCCTACGGGNGGCWGCAG-3′) and 16S_V4_R (5′-GTCTCGTGGGCTCGGAGATGTGT.ATAAGAGACAGGACTACHVGGGTATCTAATCC-3′). The amplicon library was quantified and sequenced using the MiSeq platform (Illumina, San Diego, CA, USA). After merging the reads, those shorter than 350 bp or longer than 550 bp were discarded. Chimeric sequences and singletons were trimmed using VSEARCH and the SILVA gold database. Sequencing reads with >97% similarity were clustered into operational taxonomic units using the SILVA128 database. Three samples with failed PCR and four samples with a small number of reads (<150) were removed for quality control. Finally, 111 samples were analyzed and the mean ± standard deviation valid reads were 3850.8 ± 2401.7.

### 2.4. Statistical Analysis

Biodiversity and community similarity analyses were performed using R software (Version 4.0.4.). The Wilcoxon rank-sum test and chi-square test were used to analyze the associations between continuous and categorical variables, respectively. The microbial sequences from plasma MEVs represented 30 genera and 35 species. Taxa accounting for >0.5% of the relative abundance were considered as core taxa.

Absolute OTU of each subject for both time points were combined and the average abundance of the genera was used in the analysis of pneumonia group and NHAI group.

Operational taxonomic units and the Chao1 and Shannon and Simpson indexes were used to express α-diversity. The Bray-Curtis dissimilarity index was used for principal coordinates analysis (PCoA) to assess inter-sample variation (β-diversity). We used permutational multivariate analysis of variance (PERMANOVA) to analyze β-diversity. Adonis tests from the vegan R package were performed to confirm differences between groups. Linear discriminant analysis effect size (LEfSe) was used to identify unique biomarkers based on the relative abundances of bacterial taxa. Differential taxonomic features between survivors and non-survivors by LEfSe were obtained at both day one and day seven. A diagnostic model was developed through logistic regression of biomarkers with linear discriminant analysis (LDA) scores > 3. Receiver operating characteristic (ROC) curves were used to assess the performance of MEV markers.

Tax4Fun was used for predicting the functional capabilities of microbial communities based on 16S rRNA datasets [18]. After clustering of the 16S rRNA sequencing reads, the resulting OTUs were assigned to reference sequences in the SILVA database. The SILVA -based 16S rRNA profile was transformed to a taxonomic profile of the prokaryotic KEGG organisms. The estimated taxonomic abundances were normalized by the number of 16S rRNA genes and then linked with the precomputed functional profiles of the KEGG organisms. Differences in the abundance of KEGG pathways between groups were analyzed using STAMP software (https://beikolab.cs.dal.ca/software/STAMP, accessed on 10 December 2021). Quantitative Insights in Microbial Ecology (QIIME, v. 1.8) software and the R package vegan were used for the analysis. *p* values < 0.05 were considered statistically significant.

## 3. Results

### 3.1. Demographic Characteristics of the Study Population

The 59 study participants included 41 with pneumonia and 24 at risk of NHAI. Table 1 shows the demographic characteristics of the study population. The cause of intubation, PaO_2_/FiO_2_, and C-reactive protein level differed between pneumonia and non-pneumonia groups. However, age, sex, the Charlson Comorbidity Index, and APACHE and SOFA scores were not significantly different between the groups. Patients at risk of NHAI were older and had more comorbid diseases compared to those not at risk of NHAI. However, the patient outcomes, including mortality and duration of mechanical ventilation, were not different between the groups. All patients included in this study were treated with antibiotics. Appendix A shows the antibiotic use between day one and day seven.

### 3.2. Serum MEV Diversity by Clinical Group

The participants were divided into groups based on the presence or absence of pneumonia and the NHAI risk. Mean microbiome data of days one and seven and the clinical outcomes were evaluated.

The rarefaction curve for the Chao1 index was taken to indicate the serum MEV richness. The α diversity was higher, and the slope of the rarefaction curve was steeper, in the non-pneumonia than pneumonia group (Figure 1A). The β diversity, as determined by PCoA using the Bray–Curtis distance, was significantly different between the groups (Figure 1B, *p* = 0.016).

The 28 and 29 bacterial genera accounted for more than 0.5% of the total abundance of pneumonia and non-pneumonia groups. A total of 16 bacterial genera showed significantly different compositions between the pneumonia and non-pneumonia groups (*p* < 0.05). The LEfSe showed that *Lactobacillus*, *Cutibacterium*, and *Sphingomonas* were significantly increased in the non-pneumonia compared to the pneumonia group (Figure 1C, by >1%; LDA score > 3, *p* < 0.05). The diagnostic model was based on multiple logistic regression of the microbiome markers identified through LEfSe. Model validation was performed and the area under the ROC curve (AUC) was 0.81 (Figure 1D).

The α and β diversities, determined by PCoA based on the Bray–Curtis distance, did not differ between the NHAI and non-NHAI groups, in which 30 and 29 bacterial genera accounted for >0.5% of the total abundance, respectively (Figure 2A,B). The genus-level non-NHAI diagnostic model was based on multiple regression of the biomarkers identified through LEfSe (Figure 2C). LEfSe analysis at the genus level showed that *Lactobacillus* and *Bifidobacterium* had significantly higher abundances in the non-NHAI compared to the NHAI group (Figure 2C, by >1%; LDA score > 3, *p* < 0.05). Model validation was performed and the AUC was 0.72 (Figure 2D).

### 3.3. Comparison of the Diversity of Serum MEV Composition between Survivors and Non-Survivors

The changes in α and β diversities between days one and seven did not differ in survivors and non-survivors (data not shown). The LEfSe showed that *Rhodococcus, Cloacibacterium,* and *Delftia* had significantly higher abundances on day one in non-survivors compared to survivors (Figure 3A, by >1%; LDA score > 3, *p* < 0.05). LEfSe-based plasma MEV biomarkers at day seven were *Methylobacterium* and *Acinetobacter* (Figure 3B, by >1%; LDA score > 3, *p* < 0.05). *Methylobacterium* was dominant in non-survivors and *Acinetobacter* was dominant in survivors.

In the analysis of β diversity, the Bray-Curtis dissimilarity index differed between survivors and non-survivors (Figure 3C, *p* = 0.014). Interestingly, the strains showing variation over time were identical to the day seven MEV biomarkers identified through LEfSe (*Acinetobacter* and *Methylobacterium)*. Figure 3D showed the time-dependent variation between day one and day seven in Plasma MEV. While the relative abundance of *A. radioresistens* significantly decreased only in non-survivors, the relative abundance of *Methylobacterium* significantly decreased only in survivors. At day seven, the relative abundance of *A. radioresistens* was significantly higher in survivors compared to non-survivors. In contrast, the relative abundance of *Methylobacterium* at day seven was significantly lower in survivors than non-survivors (Figure 3D). Therefore, the genus-level 28-day mortality prediction model was constructed using multiple logistic regression of the day seven biomarkers identified through LEfSe. Model validation was performed and the AUC was 0.70 (Appendix A).

### 3.4. Predicted Functions

Predictive functional profiling showed that the pneumonia group had a greater proportion of genes related to the bacterial invasion of epithelial cells and RNA transport, while the non-pneumonia group had a greater proportion of genes related to D-arginine, D-ornithine, and histidine metabolism, tuberculosis, and mannose type O glucan biosynthesis (Figure 4A).

We analyzed changes in level 3 predicted functions of MEV over time for survivors and non-survivors, respectively. The expression levels of genes related to alanine, aspartate, and glutamate metabolism, the biosynthesis of antibiotics, ribosome biogenesis in eukaryotes, and insulin resistance were increased, whereas those of genes related to the metabolism of glutathione, drugs, and xenobiotics were decreased in survivors (Figure 4B). In non-survivors, no signal pathways showed significant changes.

## 4. Discussion

The microbiome is a useful biomarker of several diseases [19,20]. Previous microbiome studies of pneumonia focused on the respiratory or gut microbiome [12,21]. Plasma EVs can disseminate to several organs and tissues through the bloodstream, mediate intercellular communication, and affect the immune system [22]. Therefore, EV metagenomics can determine the effects of bacterial infection on body systems. Blood EVs are novel diagnostic and prognostic biomarkers of ARDS, but clinical studies on EV in pneumonia are limited [23,24]. To the best of our knowledge, no previous studies have assessed the plasma MEVs in intubated patients.

Our results showed that the dominant blood microbiome phyla were *Proteobacteria*, *Firmicutes*, *Actinobacteria*, and *Bacteroidetes,* similar to previous studies [20,25]. However, unlike previous studies, the proportion of *Prooteobacteria* was <50%. Possible reasons for this discrepancy may include the sample characteristics (patients on mechanical ventilators with frequent microbial contact) or the place of residence.

In the present study, plasma MEV analysis showed that microbiome profiles vary between patient groups, such as pneumonia, non-pneumonia, NHAI, non-NHAI, survivor, and non-survivor groups. In addition, changes of specific strains in the plasma EV microbiome are related to mortality. Because all patients were treated with antibiotics after initiation of mechanical ventilation (Appendix A), it was difficult to assess the effect of antibiotics on the plasma MEV. Some previous studies on mechanically ventilated patients showed that the administration of antibiotics was not associated with a decrease in alpha–diversity of the respiratory microbiome [26,27].

In the present study, *Lactobacillus, Cutibacterium*, and *Sphingomonas* had higher abundances in the non-pneumonia compared to pneumonia group. *Cutibacterium* and *Sphingomonas* were not frequently found in the conventional cultures of respiratory samples. *Cutibacterium* is a skin commensal that can cause opportunistic infections, such as breast infection, infective endocarditis, and skin abscess [28]. *Sphingomonas* have been isolated from water, soil, and corroding copper pipes [29].

The abundances of *Lactobacillus* and *Bifidobacterium* were significantly higher in the non-NHAI compared to the NHAI group in this study. Previous studies showed that increased abundances of *Lactobacillus* and *Bifidobacterium* in pneumonia patients are associated with clinical improvement, altered gut microbiota, and a reduced incidence of ventilator-associated pneumonia [30,31]. These data suggest that *Lactobacillus* and *Bifidobacterium* regulate the behavior of other taxa and inhibit pathogen proliferation [32].

Notably, plasma MEV may predict the 28-day mortality in mechanically ventilated patients. The β diversity of MEVs significantly differed between our survivors and non-survivors. In particular, *Methylobacterium* was decreased in survivors, but not in non-survivors, whereas *A. radioresistens* was decreased in non-survivors but not in survivors. In addition, at seven days after intubation, the proportion of *Methylobacterium* was significantly higher in non-survivors than survivors, and the proportion of *A. radioresistens* was significantly higher in survivors than non-survivors.

*Methylobacterium* cause healthcare-associated infections in immunocompromised patients and are difficult to detect due to slow growth [33]. *A. radioresistens* infrequently causes infection; therefore, the details of infection with *A. radioresistens* are largely unknown [34,35]. *A. radioresistens* is resistant to radiation, desiccation, and hydrogen peroxide. In addition, it may be a hidden reservoir for carbapenem resistance, despite low expression of the ^bla^OXA-23 gene [35]. Further research is needed to determine the clinical significance of the specific strains identified in the present study.

We demonstrated the ability of plasma MEVs to predict the disease subgroups of intubated patients. Genes related to bacterial invasion of epithelial cells and RNA transport were increased in the pneumonia group. The abundances of genes related to D-arginine, D-ornithine, and histidine metabolism were significantly higher in the non-pneumonia compared to pneumonia group. D-arginine modulates the fitness of bacterial communities; enhanced metabolism of D-arginine may help to maintain polymicrobial communities by inhibiting the overgrowth of specific bacteria [36]. Histidine is a precursor of carnosine, which acts as a buffer and antioxidant [37].

Importantly, longitudinal changes in the predicted functions of MEVs differed between our survivor and non-survivor groups. It is likely that increased biosynthesis of antibiotics and ribosome biogenesis led to successful pathogen eradication and increased survival rates. Ilaiwy reported metabolic changes in sepsis, including significant enrichment of alanine, aspartate, and glutamine metabolism [38]. Our results showed that alanine, aspartate, and glutamine metabolism were increased in survivors. The effect of altered metabolism on the outcome of sepsis remains to be elucidated. The decreased metabolism of drugs and xenobiotics indicates that the inflammatory cytokines that modulate the activity of drug-metabolizing enzymes were reduced in survivors compared to non-survivors [39]. Downregulated glutathione metabolism may enhance antioxidant defense [40].

The limitations of this study included the small sample, single-center design and lack of external validation. Most of the clinical characteristics between the groups were similar, but the NHAI and non-NHAI groups showed differences in the age and number of comorbidities. In addition, it is unclear whether the predicted functions of the MEVs are the cause or consequence of disease. However, the associations between metabolic changes and the prognosis of intubated patients may provide insight into the disease pathogenesis.

In conclusion, EV-based metagenomic markers may be used to diagnose pneumonia, and for mortality risk stratification, in intubated patients. We identified associations between predicted metabolic functions and disease characteristics, such as pneumonia and 28-day survival. Large scale longitudinal studies are needed to confirm the clinical utility of EV-based markers.

## Figures and Tables

**Figure 1 jpm-12-00564-f001:**
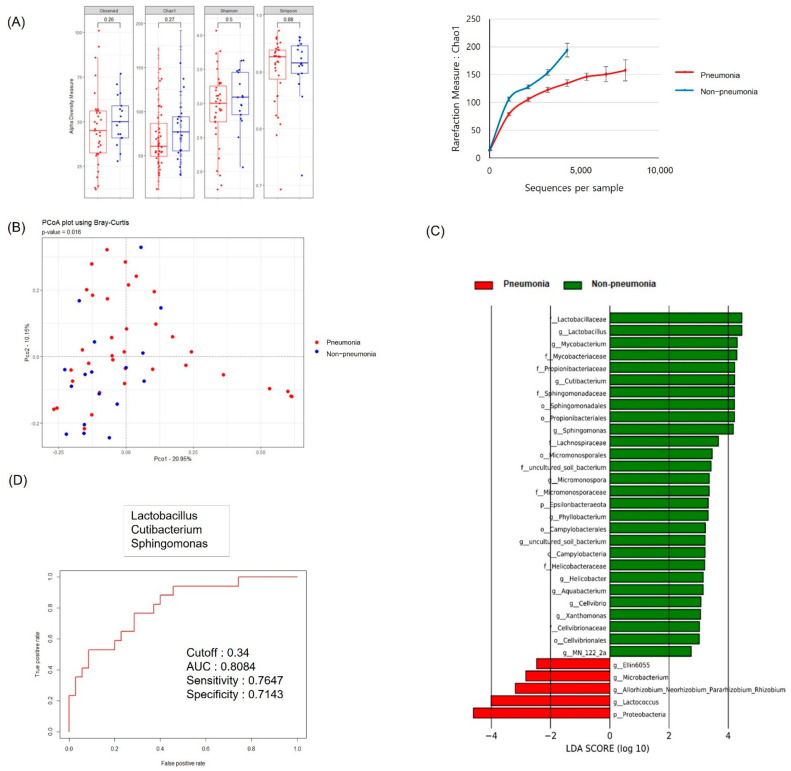
MEVs in the plasma of patients in the pneumonia and non-pneumonia groups. (**A**) α diversity measures and rarefaction curves based on the Chao1 index of species richness. (**B**) PCoA of plasma MEVs at the genus level. (**C**) LEfSe−based plasma MEV biomarkers. (**D**) ROC curves showing the accuracy of MEV strain to discriminate between pneumonia and non-pneumonia. AUC 0.81 (95% CI 0.59–0.85), *p* = 0.233. EV: extracellular vesicle; LEfSe: linear discriminant analysis effect size; ROC: receiver operating characteristic; AUC: area under the receiver operating characteristic curve; CI confidence interval; f: family; o: order; p: phylum; c: class; g: genus.

**Figure 2 jpm-12-00564-f002:**
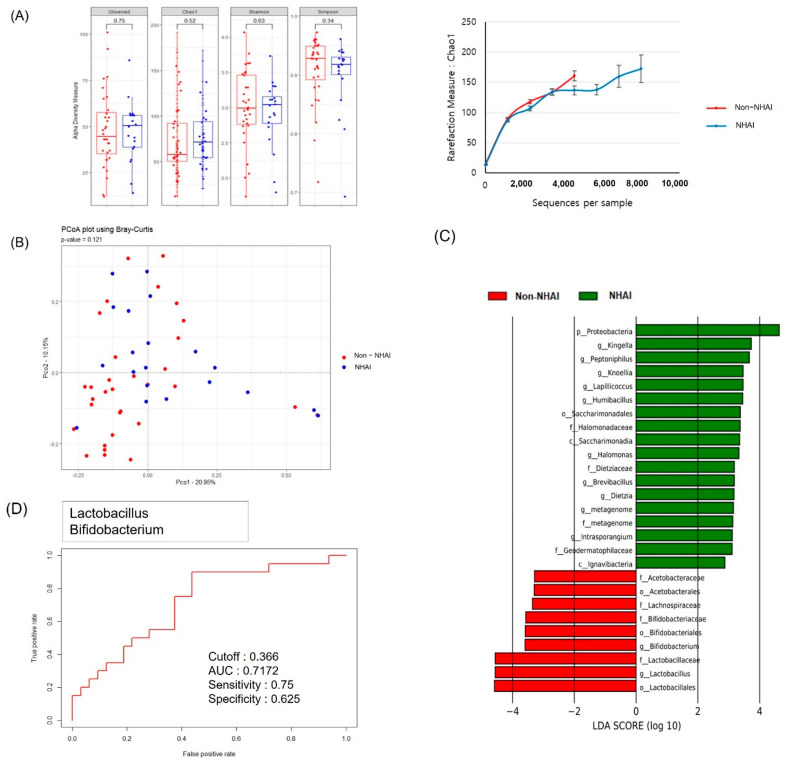
EVs in the plasma of the NHAI and non-NHAI groups. (**A**) α diversity measures and rarefaction curves based on the Chao1 index of species richness. (**B**) PCoA of plasma MEVs at the genus level. (**C**) LEfSe−based plasma MEV biomarkers. (**D**) ROC curves showing the accuracy of MEV strain to discriminate between NHAI and non-NHAI group. AUC 0.72 (95% CI 0.53,0.80) *p* = 0.240. NHAI: nursing home and hospital-associated infections; EV: extracellular vesicle; LEfSe: linear discriminant analysis effect size; ROC: receiver operating characteristic; AUC: area under the receiver operating characteristic curve. CI confidence interval; f: family; o: order; p: phylum; c: class; g: genus.

**Figure 3 jpm-12-00564-f003:**
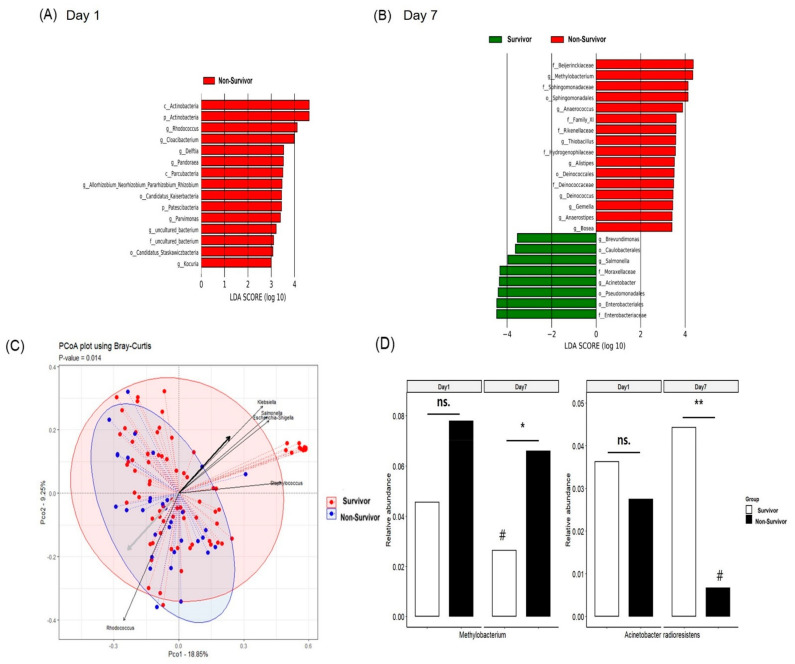
Plasma EVs of intubated patients according to 28-day mortality status. (**A**) LEfSe−based plasma MEV biomarkers at days one and (**B**) days seven (**C**) Visualization of the PCoA analysis on the Bray–Curtis dissimilarity index for the survivor group (red dashed lines) and non−survivor group (blue dashed lines). (**D**) Longitudinal changes in *Methylobacterium* and *Acinetobacter* radioresistance in the plasma EVs according to 28-day mortality status. EV: extracellular vesicle; LEfSe: linear discriminant analysis effect size; ROC: receiver operating characteristic; f: family; o: order; p: phylum; c: class; g: genus. # (day one vs. day seven, *p* < 0.05) * (survivor vs. non-survivor, *p* < 0.05), ** (survivor vs. non-survivor, *p* < 0.01).

**Figure 4 jpm-12-00564-f004:**
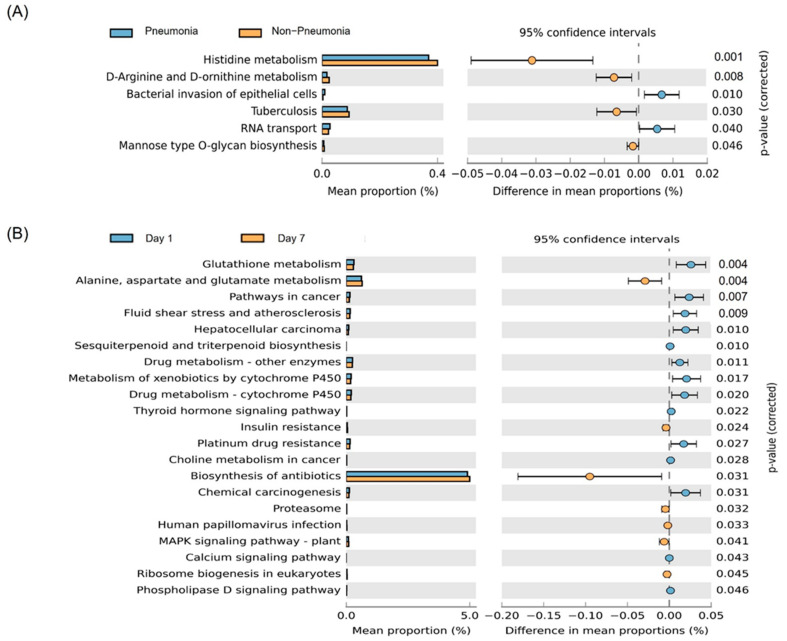
Predicted Kyoto Encyclopedia of Genes and Genomes pathways. (**A**) Level 3 functions showing differences between the pneumonia and non-pneumonia groups. (**B**) Level 3 functions showing differences over time in survivor group.

**Table 1 jpm-12-00564-t001:** Baseline characteristics of the study participants.

	Pneumonia	Non-Pneumonia	*p* Value	NHAI	Non NHAI	*p* Value
	*n* = 41	*n* = 18		*n* = 24	*n* = 35	
Age	73 (60.5, 79.5)	76 (59, 81)	0.52	77.5 (72.3, 85.0)	70 (57, 77)	0.005
Male	30 (73.2%)	12 (66.7%)	0.76	15 (62.5%)	27 (77.1%)	0.254
ARDS	8 (19.5%)	0 (0%)	0.092	4 (16.7%)	4 (11.4%)	0.704
Charlson comorbidity index	3 (1, 4)	2 (1, 2.3)	0.079	3 (2.3, 5.5)	1 (0, 2)	<0.001
Cause of intubation			<0.001			<0.001
Cardiac arrest	1 (2.4%)	3 (16.7%)		0 (0%)	4 (11.4%)	
Neurological distress	5 (12.2%)	13 (72.2%)		1 (4.2%)	17 (48.6%)	
Postoperative status	0 (0%)	1 (5.6%)		0 (0%)	1 (2.9%)	
Respiratory	35 (85.4%)	1 (5.6%)		23 (95.8%)	13 (37.1%)	
PaO_2_/FiO_2_	212.5 (133.1, 299.0)	427.4 (304.5, 463.8)	<0.001	222 (130.9, 301.0)	321 (182.8, 435.0)	0.047
Severity						
APACHE score	20 (16, 24)	22.5 (18.5, 25.3)	0.121	20.5 (16.3, 24.0)	21 (17, 25)	0.551
SOFA score	7 (6, 9)	6 (4.5, 9.3)	0.08	7 (6, 9)	7 (5, 10)	0.852
GCS	8 (6, 11)	6 (5, 8.3)	0.038	8.5 (6, 10.8)	7 (6.9)	0.241
28-day mortality	12 (29.3%)	6 (33.3%)	0.77	7 (29.2%)	11 (31.4%)	1
In-Hospital mortality	19 (46.3%)	6 (33.3%)	0.403	11 (45.8%)	14 (40.0%)	0.79
MV duration	13 (8.0, 18.0)	10 (6.8, 14.5)	0.248	13 (8, 17.5)	10 (7, 16)	0.349
CRP (mg/dL)	132.7 (45.0, 213.5)	61.6 (7.2, 135.5)	0.022	124.2 (62.8, 192.4)	96 (8, 205.6)	0.195

Data are expressed as median (interquartile range) unless otherwise indicated. NHAI: nursing home and hospital-associated infection; PaO_2_/FiO_2_: ratio of arterial oxygen partial pressure to fractional inspired oxygen; APACHE: Acute Physiology, Age, Chronic Health Evaluation II; SOFA: Sequential Organ Failure Assessment; GCS: Glasgow Coma Scale; MV: mechanical ventilation; CRP: C-reactive protein.

## Data Availability

The datasets presented in this study can be found in online repositories. The names of the repository/repositories and accession number(s) can be found at: (https://www.ncbi.nlm.nih.gov/sra/PRJNA796437, accessed on 14 January 2022).

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
