# Peer review of "Metagenomic Analysis of Plasma Microbial Extracellular Vesicles in Patients Receiving Mechanical Ventilation: A Pilot Study"

_jpm, 2022, doi:10.3390/jpm12040564_

Round 1

Reviewer 1 Report

The study by Jinkyeong Park et al tackles the study of microbiota derived EVs in the plasma of patients admited to the ICU with different diagnosis. They present a clear rationale for the chosen approach as well as the relevance of their findings. 

the study is novel and presents an interesting alternative as a prognosis marker for pneumonia. Their experimental set up is scientifically solid and presents interesting findings.

I detected major concernes regarding specific points of the study which need to be addressed before the study is ready to be published.

  1. The figures have a very low quality, they are blurry and have significant errors like cutted axes, missing legends and confusing information. The authors need to revise all their figures and improve accordingly. There is also missing information on some of the abbreviations such as the g and f in front of the different species found. 
  2. Figure 1C is not mentioned anywhere in the manuscript, this must be addresed as well and integrate it into the corresponding results section.
  3. Figure 3A, authors mentioned that Acitenobacter is dominant in the survivor group but the graph shows that other species have ahigher LDA score 
  4.  The longitudinal analysis explanation is confusing, the authors should try to improve the clarity on the 1 day and 7 days sampling and its connection to 28 days survivors or non-survivors. Sometimes the authors refer to findings at day 7 and it reads as this is the case at 28-days, it causes missinterpreation from the data. A suggestions would be to just explain once that the 28-days survivors and non-survivors as it is stated in the methods section and from then onwards just refer to these groups as survivors and non-survivors, since there isnt any additional time point evaluated
  5. I didnt completely grasped how the authors got the information on the genes for their predictive analysis, From the methods, its stated that the authors only targeted the variable 16s regions, how did your manage to quantify the expression of the different metabolic genes?, this needs to be explained.
  6. Authors present a well structured table with the different clinical parameters from the patients included in the study. However, authors should also include the antibiotic therapy these patients underwent, was it similar? this could have an effect in the MEVs the authors are measuring and could change dramatically the outcome of their predictive work.

Author Response

Dear editor

We would like to thank all of the editors and reviewers for helping us make a better revision. We revised our manuscript according to the comments and recommendations of the reviewers. We highlighted all changes in the revised manuscript in red letters.

Reviewer 1

The study by Jinkyeong Park et al tackles the study of microbiota derived EVs in the plasma of patients admited to the ICU with different diagnosis. They present a clear rationale for the chosen approach as well as the relevance of their findings.

the study is novel and presents an interesting alternative as a prognosis marker for pneumonia. Their experimental set up is scientifically solid and presents interesting findings.

I detected major concernes regarding specific points of the study which need to be addressed before the study is ready to be published.

  1. The figures have a very low quality, they are blurry and have significant errors like cutted axes, missing legends and confusing information. The authors need to revise all their figures and improve accordingly. There is also missing information on some of the abbreviations such as the g and f in front of the different species found.

We revised the figures

“Figure 1. MEVs in the plasma of patients in the pneumonia and non-pneumonia groups. (A) α diversity measures and rarefaction curves based on the Chao1 index of species richness. (B) PCoA of plasma MEVs at the genus level. (C) LEfSe-based plasma MEV biomarkers. (D) ROC curves showing the accuracy of MEV strain to discriminate between pneumonia and non-pneumonia. AUC 0.81 (95% CI 0.59-0.85), P=0.233

EV: extracellular vesicle; LEfSe: linear discriminant analysis effect size; ROC: receiver operating characteristic; AUC: area under the receiver operating characteristic curve; CI confidence interval; f: family; o: order; p: phylum; c: class; g: genus 

Figure 2. EVs in the plasma of the NHAI and non-NHAI groups. (A) α diversity measures and rarefaction curves based on the Chao1 index of species richness. (B) PCoA of plasma MEVs at the genus level. (C) LEfSe-based plasma MEV biomarkers. (D) ROC curves showing the accuracy of MEV strain to discriminate between NHAI and non-NHAI group. AUC 0.72 (95% CI 0.53,0.80) P=0.240

NHAI: nursing home and hospital-associated infections; EV: extracellular vesicle; LEfSe: linear discriminant analysis effect size; ROC: receiver operating characteristic; AUC: area under the receiver operating characteristic curve. CI confidence interval; f: family; o: order; p: phylum; c: class; g: genus 

Figure 3. Plasma EVs of intubated patients according to 28-day mortality status. (A) LEfSe-based plasma MEV biomarkers at days 1 and (B) days 7 (C) Visualization of the PCoA analysis on the Bray–Curtis dissimilarity index for the survivor group (red dashed lines) and non-survivor group (blue dashed lines). (D) Longitudinal changes in Methylobacterium and Acinetobacter radioresistance in the plasma EVs according to 28-day mortality status.

EV: extracellular vesicle; LEfSe: linear discriminant analysis effect size; ROC: receiver operating characteristic; f: family; o: order; p: phylum; c: class; g: genus. # (day 1 vs day7, P<0.05) * (survivor vs non-survivor, P<0.05)”

  1. Figure 1C is not mentioned anywhere in the manuscript, this must be addresed as well and integrate it into the corresponding results section.

“The LEfSe showed that Lactobacillus, Cutibacterium, and Sphingomonas were significantly increased in the non-pneumonia compared to pneumonia group (Figure 1C, by > 1%; LDA score > 3, p < 0.05).”

  1. Figure 3A, authors mentioned that Acitenobacter is dominant in the survivor group but the graph shows that other species have ahigher LDA score

At day 7, LEfSe showed that Acinetobacter was dominant in survivors. The related figure is figure 3B.

We revised the manuscript to avoid the confusion.

“The LEfSe showed that Rhodococcus, Cloacibacterium, and Delftia had significantly higher abundances on day 1 in non-survivors compared to survivors (Figure 3A, by > 1%; LDA score > 3, p < 0.05). LEfSe-based plasma MEV biomarkers at days 7 were Methylobacterium and Acinetobacter (Figure 3B, by > 1%; LDA score > 3, p < 0.05). Methylobacterium was dominant in non-survivors and Acinetobacter was dominant in survivors.

In the analysis of β diversity, the Bray-Curtis dissimilarity index differed between survivors and non-survivors (Figure 3C, p = 0.014). Interestingly, the strains showing variation over time were identical to the day 7 MEV biomarkers identified through LEfSe (Acinetobacter and Methylobacterium). Figure 3D showed the time-dependent variation between day 1 and day 7 in Plasma MEV. While the relative abundance of A. radioresistens significantly decreased only in non-survivors, the relative abundance of Methylobacterium significantly decreased only in survivors. At day 7, the relative abundance of A. radioresistens was significantly higher in survivors compared to non-survivors. In contrast, the relative abundance of Methylobacterium at day 7, was significantly lower in survivors than non-survivors (Figure 3D). Therefore, the genus-level 28-day mortality prediction model was constructed using multiple logistic regression of the day7 biomarkers identified through LEfSe. Model validation was performed and the AUC was 0.70 (Supplementary Figure 1).”

  1. The longitudinal analysis explanation is confusing, the authors should try to improve the clarity on the 1 day and 7 days sampling and its connection to 28 days survivors or non-survivors. Sometimes the authors refer to findings at day 7 and it reads as this is the case at 28-days, it causes missinterpreation from the data. A suggestions would be to just explain once that the 28-days survivors and non-survivors as it is stated in the methods section and from then onwards just refer to these groups as survivors and non-survivors, since there isnt any additional time point evaluated

We revised the manuscript. To avoid the misinterpretation, 28 days survivors was corrected to survivors and 28 days non-survivors to non-survivors. 모든 명칭 수정할 .

“Plasma samples were obtained 1 and 7 days after initiation of mechanical ventilation.”

“Study participants were divided into survivors and non-survivors based on the 28-day all-cause mortality.”

“Differential taxonomic features between survivors and non-survivors by LEfSe were obtained at both day1 and day7.”

  1. I didnt completely grasped how the authors got the information on the genes for their predictive analysis, From the methods, its stated that the authors only targeted the variable 16s regions, how did your manage to quantify the expression of the different metabolic genes?, this needs to be explained.

We revised the manuscript to explain in detail how we predicted functional profiles from metagenomic 16S rRNA data

“Tax4Fun was used for predicting the functional capabilities of microbial communities based on 16S rRNA datasets [18]. After clustering of the 16S rRNA sequencing reads, the resulting OTUs were assigned to reference sequences in the SILVA database. The SILVA -based 16S rRNA profile was transformed to a taxonomic profile of the prokaryotic KEGG organisms. The estimated taxonomic abundances were normalized by the number of 16S rRNA genes and then linked with the precomputed functional profiles of the KEGG organisms. Differences in the abundance of KEGG pathways between groups were analyzed using STAMP software (http://kiwi.cs.dal.ca/Software/STAMP).”

  1. Authors present a well structured table with the different clinical parameters from the patients included in the study. However, authors should also include the antibiotic therapy these patients underwent, was it similar? this could have an effect in the MEVs the authors are measuring and could change dramatically the outcome of their predictive work.

All patients included in this study were treated with antibiotics. Therefore, it is difficult to compare the differences in microbiome according to the presence of antibiotics.

Antibiotic types are summarized in the table below.

Antibiotic used
between day 1 and day 7

Pneumonia

Non-pneumonia

P

NHAI

Non NHAI

P

Survivor

Non-survivor

P

n=41

n=18

n=24

n=35

n=41

n=18

b-lactams (only)

5

(12.2%)

8

(44.4%)

0.012

1

(4.2%)

12 (34.3%)

0.01

9

(22%)

4

(22.2%)

0.431

b-lactams plus fluoroquinolones

6

(14.6%)

3

(16.7%)

2

(8.3%)

7

(20%)

8 (19.5%)

 1

(5.6%)

b-lactam plus vancomycin

19

(46.3%)

7

(38.9%)

14 (58.3%)

12 (34.3%)

18 (43.9%)

8

(44.4%)

b-lactam plus vancomycin plus
other antibiotics

 (colistin or fluoroquinolones)

11

(26.8%)

0

(0%)

7 (29.2%)

4 (11.4%)

6 (14.6%)

5

(27.8%)

One previous study on mechanically ventilated patients showed that administration of antibiotics was not associated with a decrease in alpha –diversity of respiratory microbiome. The authors stated the influence of antibiotics was not as overwhelming as might be expected.

Zakharkina, T.; Martin-Loeches, I.; Matamoros, S.; Povoa, P.; Torres, A.; Kastelijn, J.B.; Hofstra, J.J.; de Wever, B.; de Jong, M.; Schultz, M.J., et al. The dynamics of the pulmonary microbiome during mechanical ventilation in the intensive care unit and the association with occurrence of pneumonia. Thorax 2017, 72, 803-810, doi:10.1136/thoraxjnl-2016-209158.

 Similarly, the study of Anais et al discarded a major effect of antibiotic therapy on the lung microbiome signature of patients with IPA (the proportion of intubated patients was 22%).

Herivaux, A.; Willis, J.R.; Mercier, T.; Lagrou, K.; Goncalves, S.M.; Goncales, R.A.; Maertens, J.; Carvalho, A.; Gabaldon, T.; Cunha, C. Lung microbiota predict invasive pulmonary aspergillosis and its outcome in immunocompromised patients. Thorax 2022, 77, 283-291, doi:10.1136/thoraxjnl-2020-216179.

Thank you very much for your insightful and thorough advice and provisional acceptance of the manuscript.

Sincerely yours,

Reviewer 2 Report

Although the sample size was small and single-center, as indicated by the authors in the limitations, it is a well-designed prospective study with solid conclusions. Nevertheless, the authors should explain what bias reduction measures were introduced to mitigate the fact that the Charlson index in patients with pneumonia and nonpneumonic infection is (significantly) higher than in the control group.

Author Response

Dear editor

We would like to thank all of the editors and reviewers for helping us make a better revision. We revised our manuscript according to the comments and recommendations of the reviewers. We highlighted all changes in the revised manuscript in red letters.

Reviewer 2

Although the sample size was small and single-center, as indicated by the authors in the limitations, it is a well-designed prospective study with solid conclusions. Nevertheless, the authors should explain what bias reduction measures were introduced to mitigate the fact that the Charlson index in patients with pneumonia and nonpneumonic infection is (significantly) higher than in the control group.

This study included 59 patients that received mechanical ventilation within 48 h of admission to the ICU of Chuncheon Sacred Heart Hospital. The control group as patients without intubation were not included in the study. We compared the study participants by categorizing them into pneumonia and non-pneumonia, NHAI group and non-NHAI group and survivors and non-survivors.

As in the table below, there was no significant difference in the Charlson comorbidity index, age or ARDS between pneumonia and non-pneumonia. Therefore, we did not perform adjustment for variables.

Pneumonia

Non-pneumonia

P value

NHAI

Non NHAI

P value

n=41

n=18

n=24

n=35

Age

73 (60.5,79.5)

76 (59,81)

0.52

77.5 (72.3,85.0)

70 (57,77)

0.005

Male

30 (73.2%)

12 (66.7%)

0.76

15 (62.5%)

27 (77.1%)

0.254

ARDS

8 (19.5%)

0 (0%)

0.092

4 (16.7%)

4 (11.4%)

0.704

Charlson comorbidity index

3 (1,4)

2(1,2.3)

0.079

3 (2.3,5.5)

1 (0,2)

<0.001

Cause of intubation

<0.001

<0.001

Cardiac arrest

1 (2.4%)

3 (16.7%)

0 (0%)

4 (11.4%)

Neurological distress

5 (12.2%)

13 (72.2%)

1 (4.2%)

17 (48.6%)

Postoperative status

0 (0%)

1 (5.6%)

0 (0%)

1 (2.9%)

Respiratory

35 (85.4%)

1 (5.6%)

23 (95.8%)

13 (37.1%)

PaO2/FiO2

212.5 (133.1,299.0)

427.4 (304.5,463.8)

<0.001

222 (130.9,301.0)

321 (182.8,435.0)

0.047

Severity

APACHE score

20 (16,24)

22.5(18.5.25.3)

0.121

20.5 (16.3, 24.0)

21 (17,25)

0.551

SOFA score

7 (6,9)

6 (4.5,9.3)

0.08

7 (6, 9)

7 (5,10)

0.852

GCS*

8 (6,11)

6(5,8.3)

0.038

8.5 (6,10.8)

7 (6.9)

0.241

28‐day mortality

12(29.3%)

 6 (33.3%)

0.77

7 (29.2%)

11 (31.4%)

1

In-Hospital mortality

19 (46.3%)

6 (33.3%)

0.403

11 (45.8%)

14 (40.0%)

0.79

MV duration

13 (8.0,18.0)

10 (6.8,14.5)

0.248

13 (8,17.5)

10 (7,16)

0.349

CRP (mg/dL)

132.7 (45.0,213.5)

61.6 (7.2,135.5)

0.022

124.2 (62.8,192.4)

96 (8,205.6)

0.195

Survivor

Non-survivor

P value

n=41

n=18

Age

72 (58.5,78.5)

76.5 (66.8,85.3)

0.103

Male

28 (68.3%)

14 (77.8%)

0.545

ARDS

4 (9.8%)

4 (22.2%)

0.231

Charlson comorbidity index

2 (1, 4)

2 (0.8,3)

0.312

Cause of intubation

0.748

Cardiac arrest

2 (4.9%)

2 (11.1%)

Neurological distress

13 (31.7%)

5 (27.8%)

Postoperative status

1 (2.4%)

 0 (0%)

Respiratory

25 (61%)

11 (61.1%)

PaO2/FiO2

242 (160.3,367.5)

293.5(157.2,434.3)

0.548

Severity

APACHE score

20 (16,24)

23 (19.8,26.3)

0.064

SOFA score

7(5,9)

7.5 (6,9.3)

0.239

GCS*

8 (6,10)

6 (5,11)

0.237

MV duration

10 (7,15)

14 (8,19)

0.541

CRP (mg/dL)

115 (45.3,219.1)

74.2 (9.7,138.7)

0.064

Similarly, when comparing survivors and non-survivors, several variables did not differ between the two groups.

However, age and Charlson comorbidity index were higher in NHAI group compared with non-NHAI group.  We added this in the Limitation section

“The limitations of this study included the small sample, single-center design and lack of external validation. Most of the clinical characteristics between the groups were similar, but the NHAI and non-NHAI groups showed difference in age and number of comorbidities. In addition, it is unclear whether the predicted functions of the MEVs are the cause or consequence of disease. However, the associations between metabolic changes and the prognosis of intubated patients may provide insight into the disease pathogenesis.”

Thank you very much for your insightful and thorough advice and provisional acceptance of the manuscript.

Sincerely yours,

Round 2

Reviewer 1 Report

the authors have addressed all my comment and the manuscript has improved considerably, Authors should include the table with antibiotic treatments as supplementary for other interested parties as I believe this is important information in the context of the study.

Author Response

We included the supplementary table 1.

Supplementary table S1. Antibiotic therapy per group between day 1 and day7

Antibiotic used
between day 1 and day 7

Pneumonia

Non-pneumonia

P

NHAI

Non NHAI

P

Survivor

Non-survivor

P

n=41

n=18

n=24

n=35

n=41

n=18

b-lactams (only)

5

(12.2%)

8

(44.4%)

0.012

1

(4.2%)

12 (34.3%)

0.01

9

(22%)

4

(22.2%)

0.431

b-lactams plus fluoroquinolones

6

(14.6%)

3

(16.7%)

2

(8.3%)

7

(20%)

8 (19.5%)

 1

(5.6%)

b-lactam plus vancomycin

19

(46.3%)

7

(38.9%)

14 (58.3%)

12 (34.3%)

18 (43.9%)

8

(44.4%)

b-lactam plus vancomycin plus
other antibiotics

 (colistin or fluoroquinolones)

11

(26.8%)

0

(0%)

7 (29.2%)

4 (11.4%)

6 (14.6%)

5

(27.8%)

NHAI: nursing home and hospital-associated infection;

All patients included in this study were treated with antibiotics. Supplementary table 1 shows the antibiotic use between day 1 and day 7.

In the present study, plasma MEV analysis showed that microbiome profiles vary between patient groups, such as pneumonia, non-pneumonia, NHAI, non-NHAI, survivor, and non-survivor groups. In addition, changes of specific strains in plasma EV microbiome are related to mortality. Because all patients were treated with antibiotics after initiation of mechanical ventilation (Supplementary Table S1), it was difficult to assess the effect of antibiotics on the plasma MEV. Some previous studies on mechanically ventilated patients showed that administration of antibiotics was not associated with a decrease in alpha –diversity of respiratory microbiome[26,27].

Thank you very much for your insightful and thorough advice and provisional acceptance of the manuscript.

Sincerely yours,
